# Optimal Levels of Fish Meal and Methionine in Diets for Juvenile *Litopenaeus vannamei* to Support Maximum Growth Performance with Economic Efficiency

**DOI:** 10.3390/ani13010020

**Published:** 2022-12-21

**Authors:** Alberto J. P. Nunes, Karthik Masagounder

**Affiliations:** 1LABOMAR—Instituto de Ciências do Mar, Universidade Federal do Ceará, Avenida da Abolição, 3207, Meireles, Fortaleza 60165-081, Ceará, Brazil; 2Evonik Operations GmbH, 10-B227, Rodenbacher Chausse 4, 63457 Hanau, Germany

**Keywords:** fish meal, replacement, methionine, DL-methionyl-DL-methionine, supplementation

## Abstract

**Simple Summary:**

The shrimp feed industry is constantly looking for opportunities to minimize the dependency on expensive fish meal and to keep the industry profitable and sustainable. When fish meal is replaced with alternative protein sources available today, methionine (Met) is often the first limiting amino acid. This work investigated the optimal levels of fish meal (FML) and dietary Met required to optimize growth performance of juvenile *Litopenaeus vannamei* with economic efficiency. The study involved two feeding trials, one with outdoor tanks (1 m^3^ volume) to evaluate shrimp growth performance over a 70-day feeding period and the other with indoor tanks (60 L) to evaluate feed digestibility over a 93-day period. The study used DL-methionyl-DL-methionine as the supplemental Met source. Under 0 and 6% FML conditions, total dietary Met levels of 0.69 and 0.82%, respectively, were required to maximize shrimp performance. In comparison, at 12 and 18% FML, a dietary Met content of only 0.58% was sufficient. Diets with 0 FML or with only 6% delivered the highest profit and return on investment compared to diets with higher levels. Overall, results indicated FML in shrimp feeds can be minimized or eliminated without impairing growth performance, providing Met requirement is met with appropriate sources.

**Abstract:**

This work investigated the optimal levels of fish meal (FML) and dietary methionine (Met) required for maximum growth performance of juvenile *Litopenaeus vannamei* with economic efficiency. Four sets of diets were prepared to contain 0.00, 6.00, 12.00 and 18.00% FML. Each set was supplemented with DL-methionyl-DL-methionine (DL-Met-Met) to result in a total dietary Met (Met + Cys) content of 0.58 (1.05), 0.69 (1.16), and 0.82% (1.29%), on a fed basis. Shrimp of 1.00 ± 0.08 g were stocked in 60 outdoor tanks of 1 m^3^ with 100 shrimp/m^2^, allowing five replications per dietary group. Shrimp in all the groups were fed 10 times daily for 70 days. In a subsequent trial, dietary protein and amino acid digestibility of four FML groups, but only at high dietary Met levels (~0.82%), were evaluated in 40 60 L indoor tanks (11 replicates per diet) for 93 days with 70 shrimp/m^2^. Final shrimp survival (92.85 ± 4.82%, mean ± standard deviation), weekly weight gain (1.17 ± 0.08 g), apparent feed intake (13.3 ± 0.5 g of feed per stocked shrimp), and feed conversion ratio (1.18 ± 0.06) were unaffected by dietary FML level and Met content. Gained yield was adversely affected when FML was reduced from 18 and 12% (1156 and 1167 g/m^2^, respectively) to 0 (1090 g/m^2^), but no change was observed at 6% (1121 g/m^2^). A significant interaction was detected between FML level and dietary Met. Under 0 and 6% FML conditions, higher levels of total dietary Met, 0.69 and 0.82%, respectively, were required to maximize shrimp BW. In comparison, at 12 and 18% FML, a dietary Met content of only 0.58% was sufficient. Overall, results indicated the use of FML can be minimized or completely eliminated without major detrimental effects on feed digestibility or shrimp growth performance, as long as proper supplementation of Met is carried out. Diets with 0 FML or with only 6% delivered the highest profit and return on investment compared to diets with higher levels.

## 1. Introduction

Industrially compounded shrimp feeds are among the largest global consumers of fish meal within the aquaculture industry [1,2,3,4]. As a result, replacement of fish meal by other proteins has been the subject of a number of publications. Most studies have supported that dietary fish meal levels can be significantly reduced without adverse effects on shrimp growth performance. Consequently, there has been a progressive decrease in the inclusion of fish meal in shrimp feeds, from more than 25% in the 1990s [1,5] to 12% or less in the past decade [1,4]. This has led to a corresponding decline in the fish-in-fish-out (FIFO) ratio [2] from an estimated 2.81 in 2007 [5] to 0.82 in 2017 [4]. Further reduction or complete withdrawal of fish meal from shrimp feeds has been achieved under experimental culture conditions [6,7,8,9,10,11]. However, effective replacement levels vary according to culture conditions, substitute protein, feed formulation, and sources of supplemental amino acids. 

The most common proteins adopted to replace fish meal in shrimp feeds are commodity by-products from the animal slaughtering industry (poultry by-product meal [11,12,13,14]; meat and bone meal [12,13,15]; porcine meat meal [16]) and agriculture (soybean meal [6,7,17,18,19,20,21,22,23,24]; soy protein concentrate [11,13,21,25,26,27]; canola meal [18,28,29]; corn meal [30]; cottonseed meal [31]; peanut meal [10,19]). Unconventional protein ingredients have also been evaluated with promising results (bacterial meal [32,33,34]; biofloc meal [26,35,36]; earthworm [37]; insect meal [38]; microalgae meal [39,40]). Soybean meal is by far the most common and preferred ingredient to support fish meal replacement in practical shrimp feeds due to its year-round availability and competitive and less volatile prices. Additionally, most studies have used a combination of protein sources to replace fish meal. Regardless of the protein sources chosen, studies have shown that formulation of low fish meal diets relies on a balanced supplementation of essential amino acids [20,27,41,42], fatty acids [27], and feed attractants [9,28,29,41].

Methionine (Met) is considered the most impacted essential amino acid (EAA) when fish meal is challenged. Recommended dietary Met levels in shrimp feeds have ranged between 0.7 and 1.0% of the diet (as-fed basis) depending on shrimp species (*Penaeus monodon* [43], *Marsupenaeus japonicus* [44]), source of supplemental Met [13,20,45,46,47,48], growth stage [49], culture conditions (stocking density [50], water exchange regime [51], feed allowance [52]), and dietary protein level [53,54]. However, very little information is available on the optimal dietary Met levels in response to graded levels of fish meal in order to support maximum shrimp growth performance with economic efficiency. The present work evaluated the growth performance of juvenile *Litopenaeus vannamei*, the feed digestibility, and the economics of fish meal (FML) reduction with the dietary supplementation of DL-methionyl-DL-methionine (DL-Met-Met) under intensive culture conditions. 

## 2. Material and Methods

### 2.1. Rearing System and Water Preparation 

This study consisted of two separate experimental stages. The 1st stage was designed to evaluate shrimp growth performance fed graded levels of FML and Met. A 2nd stage determined the apparent digestibility coefficients (ADCs) for crude protein (ACPDCs) and amino acids (AAADCs) of diets with graded levels of FML using a fixed dietary Met content.

For the growth performance evaluation, an outdoor rearing system as described by Façanha et al. [50,51,52] and Nunes et al. [54] was used. The system consisted of independent 1.0 m^3^ outdoor tanks (1.02 m^2^ bottom area × 0.74 m height), each equipped with a perforated lid on the top to avoid shrimp from escaping. The system operated with a continuous water recirculation at a rate of 100 mL/second (14.4% a day). Effluent water was drained into a 10 m^3^ sump with constant aeration which pumped surface water back to two header tanks of 20 m^3^ each. Water was then continuously distributed to rearing tanks with a 3-hp pump. No mechanical filtration was carried out in culture water during the study period. Water was fertilized prior to shrimp stocking by applying liquid sugarcane molasses along with a 500 µm shrimp feed (minimum of 35% crude protein, CP) at a 1:1 ratio (20 g/m^3^, as-is basis) over a five-day period. Culture water was then allowed to mix with strong aeration for three additional days before shrimp stocking.

The in vivo digestibility assay was carried out in rectangular 61 L indoor tanks (31.0 × 35.5 × 55.5 cm, height × width × length; bottom area of 0.19 m^2^), each equipped with its own water inlet and outlet, aeration, and feeding tray. The rearing system was the same as described by Sabry-Neto et al. [55] and Vieira et al. [56]. It was operated with a continuous water recirculation regime at a rate between 15.65 and 21.81 L/h (26.1–36.4% of the tank volume/h). All culture water was chemically disinfected and filtered prior to shrimp stocking.

### 2.2. Shrimp Stocking

The shrimp species used in this study was the Pacific whiteleg shrimp, *L. vannamei*, purchased as post-larvae (PLs) from a commercial hatchery (Aquatec Aquacultura Ltda., Canguaretama, Brazil). A batch of 110,000 PL8 (245 PLs/g) was transported to the lab in 11 double-plastic sealed bags at a density of 667 animals/L (10,000 PLs/bag) under 35 g/L salinity, 24 °C temperature, 8.09 pH, and 148 mg/L CaCO_3_ alkalinity conditions. At arrival, PLs were acclimated and stocked in five cylindrical nursery tanks of 23.85 m^3^ each at about 22,000 animals per tank or 1 PL/L. Shrimp were fed a 40% CP commercial crumbled feed and nursery-reared until juvenile stage. In the end of the nursery stage, shrimp were size-graded to homogenize body weight (BW). Culling was carried out by weighing shrimp individually in a 0.01 g precision scale. A total of 6120 shrimp of 1.00 ± 0.08 g (mean ± standard deviation) were transferred to 60 outdoor tanks (five replicate tanks per diet) and stocked with 100 animals/m^2^ (102 shrimp/tank) for the growth performance trial. Later, a total of 600 shrimp of 4.47 ± 0.14 g BW were stocked in 44 tanks (11 replicate tanks per diet) with 70 animals/m^2^ (15 shrimp/tank) for the digestibility assay.

### 2.3. Experimental Diets

For the growth performance evaluation, three sets of pelleted diets were prepared, each containing three levels of total Met with four inclusion levels of FML (12 diets in total). Each set of diets was supplemented with DL-methionyl-DL-methionine (AQUAVI^®^ DL-Met-Met, Evonik Operations GmbH, Hanau, Germany) to achieve the following mean (±standard deviation, SD) total dietary Met content (as-is basis): 0.58 ± 0.02%, (0.58% Met) 0.69 ± 0.02 (0.69% Met), and 0.82 ± 0.02% (0.82% Met), with a corresponding Met + Cys (cysteine) content of 1.05 ± 0.01, 1.16 ± 0.03, and 1.29 ± 0.02%, respectively (Table 1).

The dietary inclusions of FML ranged from a maximum of 18.00% (18% FML) to 12.00% (12% FML), 6.00% (6% FML), and no FML at all (0% FML). FML was replaced for soybean meal (SBM) which increased from a minimum of 20.73 (diet 18% FML) to 28.95 (12% FML), 37.8 (6% FML), and 46.08% (0% FML), respectively. All diets required DL-Met-Met supplementation to meet the targeted Met levels, except diets with 18.00% FML and a final total Met content of 0.58%. In this case, Met was only derived from protein-bound sources. 

Diets reached a mean CP and total lipid content of 34.57 ± 0.37% and 7.73 ± 0.44% on a fed basis, respectively (Table 2). To maximize protein utilization and keep a well-balanced amino acid (AA) profile, diets were formulated on an ideal protein basis using lysine (Lys) as the first limiting and reference AA [57]. Total dietary Lys, threonine (Thr), and arginine (Arg) content reached 1.78 ± 0.04, 1.30 ± 0.02, and 2.25 ± 0.15% (as-is), respectively. A commercial grower shrimp feed with 39.25% CP and 6.90% total lipids was used as a reference (CTL). Total Met (Met + Cys), Lys, Thr, and Arg in the CTL diet were 0.86 (1.29), 2.21, 1.41, and 2.30%, respectively. Crude ash content was 12.17%, equivalent to the experimental diets containing the highest level of FML, i.e., 18%. However, total fiber content was 2.99%, higher than experimental diets with 46.08% SBM. 

For the digestibility assay, a new set of four diets containing 0, 6, 12, and 18% FML and a constant level of total dietary Met was formulated (Table 3). Formulas were similar to the ones used in the growth performance evaluation but included 1.00% of chromic oxide to act as an inert marker. DL-Met-Met was supplemented at 0.41, 0.34, 0.27, and 0.20% (as-is) in diets with 0, 6, 12, and 18% FML, respectively. Finished diets reached a total CP, Met (Met + Cys), Lys, Thr, and Arg content of 33.99 ± 0.33, 0.81 ± 0.01 (1.27 ± 0.03), 1.73 ± 0.02, 1.24 ± 0.01, and 1.99 ± 0.02% (as-is), respectively (Table 4).

All experimental diets were produced using a laboratory extruder following the methodology described by Nunes et al. [58]. Finished diets (pellets of 2.0 mm in diameter by 5 mm in length) were stored at 16 °C in sealed containers. For diets used in the growth performance, physical water stability measured by the orbital shaker method was 87.44 ± 2.23% (*n* = 75) for the treatments and 93.14 ± 0.32% (*n* = 5) for the CTL. 

### 2.4. Shrimp Feeding, Feces Collection, and Water Quality

#### 2.4.1. Growth Performance Trial

In outdoor tanks, shrimp were fed 10 times during daylight using an automatic feeder [59]. The daily rations were calculated based on the equation MM = 0.0931BW^0.6200^, where MM is the maximum amount of feed that can be consumed daily by an individual with a specific BW [60]. Previous work had shown that the MM can be restricted by 28.8% without any detrimental effect on shrimp growth performance [61]. Therefore, the daily meals were reduced by 30% across all treatments to control feed conversion ratio (FCR). Meals were adjusted daily, assuming a fixed weekly drop in shrimp survival (by 0.38%) and a BW gain of 100 mg/shrimp/day. Starting from the 11th day of rearing, every two weeks, 10 shrimp/tank were sampled to determine their mean BW gain. Until the next sampling, meals were adjusted assuming an average daily weight gain achieved in the previous week for each specific rearing tank, maintaining a fixed 0.38% weekly drop in shrimp survival across all diets. No feed leftovers were collected during the rearing period. Dead animals were not replaced throughout the culture period.

#### 2.4.2. Digestibility

In the digestibility assay, shrimp were fed in excess at 07:00 a.m., 01:00, and 04:00 p.m. Diets were delivered manually and exclusively in feeding trays (diameter of 9.75 cm). Daily meals were adjusted according to the amount of feed leftovers collected from feeding trays, which, when present, were collected, dried in a convection oven, weighed, and discarded. Feces were collected by siphoning four times daily [56]. Feces samples were gently rinsed with distilled water for salt removal and stored at −23 °C. All samples were freeze-dried prior to chemical analysis.

Water pH, temperature, and salinity were measured once daily starting at 09:00 a.m. in all tanks. In outdoor tanks, mean values reached 7.87 ± 0.20 (7.00–8.90, *n* = 4452), 28.5 ± 0.6 °C (*n* = 4452), and 39 ± 3 g/L (*n* = 4452), respectively. In the indoor tanks, mean pH, temperature, and salinity reached 7.81 ± 0.29 (*n* = 3468), 30.6 ± 0.9 °C (*n* = 3400), and 39 ± 4 g/L (*n* = 3264), respectively. In both rearing systems, dissolved oxygen was kept saturated over the complete rearing period. 

### 2.5. Chemical and Physical Analyses of Feeds and Feces

Feed chemical analyses followed standard methods [62]. DM was determined in a convection oven for 24 h at 105 °C. CP was analyzed with the Kjeldahl method of nitrogen estimation [62]. Ash content was determined by burning samples in a muffle at 600 °C for 2 h (AOAC 942.05) and crude fiber by enzymatic-gravimetric determination (AOAC 992.16). Amino acid content in diets and feces samples was analyzed with wet chemistry (AMINOLab^®^, Evonik Operations GmbH, Hanau, Germany) using ion exchange chromatography [63,64]. Experimental diets and the CTL feed were analyzed for physical stability in seawater (35 g/L salinity) using the orbital shaker method described by Nunes et al. [59]. Readings were determined for five replicate samples of each diet and CTL. Chromium oxide content in shrimp feces and experimental diets used to determine digestibility was determined in duplicate using electrothermal atomic absorption spectrometry (ETAAS) by SGS Analytics Germany GmbH (Jena, Germany).

### 2.6. Shrimp Growth Performance 

Shrimp were reared for 70 and 93 days in outdoor and indoor tanks, respectively. At harvest, all live shrimp were collected, counted, and individually weighed to a 0.01 g precision scale. Final shrimp survival (S, %) was calculated as S = (POPf/POPi) × 100, where POPi = number of stocked shrimp and POPf = number of shrimp at harvest. The weekly weight gain (WWG, g/week) was determined by the formula: WWG = [(BWf − Bwi)/t] × 7, where Bwi = wet shrimp body weight (BW, g) at stocking, BWf = final shrimp BW at harvest, and t = number of days in culture. The gain in shrimp yield (YIE, g of shrimp biomass gained/m^2^) was determined as YIE = (BIOf − BIOi) ÷ tank bottom area (m^2^), where BIOi = initial shrimp biomass (g) per tank, BIOf = final shrimp biomass (g) per tank, and tank bottom area = 1.02 and 0.19 m^2^ (outdoor and indoor tanks, respectively). FCR was calculated on a DM basis, by dividing the total inputs of feed (g, dry-matter basis, DM) delivered during the entire rearing period by the total harvested shrimp biomass (g, as-is basis) from each tank. The apparent feed intake (AFI, g of feed delivered divided by the number of stocked shrimp) was calculated by dividing the total amount of feed delivered (g, DM basis) by the number of stocked shrimp.

### 2.7. In Vivo Digestibility

The concentration of Cr_2_O_3_ in the finished diets and in shrimp feces was used to determine the ADC, according to the formula (Cho et al. 1982):(1)$$\mathrm{ADC}=100\;-\left[100\left(\frac{{\%\mathrm{Cr}}_{2}O_{3}d\;}{{\%\mathrm{Cr}}_{2}O_{3}f}\right)\times \left(\frac{\%\mathrm{Nf}\;}{\%\mathrm{Nd}}\right)\right]$$
where, ADC = apparent digestibility coefficient of CP (in %) and (or) AA (in %); Cr_2_O_3_d = concentration (in %) of chromic oxide in the diet; Cr_2_O_3_f = concentration (in %) of chromic oxide in shrimp feces; Nd = concentration (in %) of CP and AA in the diets; Nf = concentration (in %) of CP and AA in shrimp feces.

### 2.8. Economic Analysis

The cost of formulation of each individual diet was first calculated by using local market prices of each ingredient and feed additives (Table 1). The price of FML, SBM, and DL-Met-Met were USD 1.300, 0.415, and 5.000 per kg, respectively. Feed sale price was calculated by increasing 30% of the formula costs to account for feed mill margins, manufacturing, packaging, marketing, and other miscellaneous costs involved in feed production and sales. Feeds accounted for 40% of the total shrimp production costs (USD/kg) with the remainder attributed to other variable (PLs, amendments, sediment removal, electricity, fuel, labor) and fixed costs [65]. The total production cost (USD/kg) was determined by multiplying the feed sale price (USD/kg) by the FCR and the gained shrimp yield (kg). The farm gate price for shell-on, head-on shrimp was estimated at USD 3.42/kg. An excess of USD 0.19/kg was added on top of the final price for every one gram of shrimp in excess of 10 g BW at harvest (Brazilian Shrimp Farmers Association, February, 2022). The gross revenue (USD/kg) was determined by multiplying the farm gate shrimp price (USD/kg) with the gained shrimp yield (kg) from each tank. The gross profit (USD/kg) was given as the gross revenue subtracted by the total production cost. The return on investment (ROI, %) was calculated by subtracting the gross revenue by the total production cost and then dividing the result by the total production cost multiplied by 100. 

### 2.9. Statistical Analysis

One-way and two-way analyses of variance (ANOVAs) were used to compare the means of shrimp growth and economic performance as a function of the dietary FML inclusion and (or) Met content. One-way ANOVA was used to compare the means of economic efficiency. When significant differences were detected, they were compared two-by-two with Tukey’s HSD test. The significant level of 5% was set in all statistical analyses. The statistical package IBM^®^ SPSS^®^ Statistics 23.0 (SPSS Inc., Chicago, IL, USA) was used.

## 3. Results

### 3.1. Growth Performance

In outdoor tanks, shrimp final survival was high (92.7 ± 4.7%) and unaffected by dietary Met content, inclusion level of FML, or their interaction (*p* > 0.05, Table 5). Additionally, final survival did not differ between shrimp fed the experimental diets and the CTL (89.7 ± 2.8%). Gained shrimp yield increased progressively with higher levels of FML, from a low of 1090 ± 54 (0% FML) to a high of 1166 ± 66 g/m^2^ (12% FML). However, yield did not differ statistically when shrimp were fed diets with 18, 12, or 6% FML, regardless of the dietary Met content. The elimination of FML did not affect yield when compared to shrimp fed 6% FML (1121 ± 68 g/m^2^), but it was significantly lower than those fed 12 and 18% FML. The increase in the dietary Met levels from 0.58 (1127 ± 56 g/m^2^) to 0.69 (1145 ± 45 g/m^2^) or 0.82% (1135 ± 84 g/m^2^) had no statistical effect on gained yield or a significant interaction with FML level. Shrimp fed the CTL (1040 ± 33 g/m^2^) achieved a lower yield compared to those fed diets containing 0.69% Met at all levels of FML, except the highest (18%). At 0.58% Met, higher FML levels were required (12 and 18% FML) to significantly increase yield beyond the CTL. In comparison, at 0.82% Met, there was no significant difference in yield between shrimp fed the CTL and the experimental diets, except when 18% FML was used. In this case, yield was higher for the latter compared to the CTL. 

Shrimp weekly growth rate exceeded 1.1 g regardless of the dietary treatment. No difference or significant interaction was detected in growth between shrimp fed different levels of dietary Met and/or FML (*p* > 0.05). Shrimp fed experimental diets also grew at a similar rate compared to those fed the CTL. Apparent feed intake (AFI) did not differ statistically as a result of FML level or dietary Met content (*p* > 0.05). However, some of the experimental diets showed a higher AFI than the CTL. FCR was generally low, between 1.15 to 1.22. FCR was also not affected by variations in FML or dietary Met. FCR for experimental diets was within the range of the CTL (1.23 ± 0.02). Exceptions were diets with 6% FML containing 0.69% Met and 18% FML with 0.58% Met, which resulted in statistically lower values compared to the CTL. 

Final shrimp BW at harvest ranged between 12.61 ± 0.92 (0% FML and 0.58% Met) and 13.92 ± 0.51 g (12% FML and 0.58% Met; Figure 1). Final BW was affected by both FML levels and dietary Met content, with a strong interaction between the two (*p* < 0.0001). With 0 and 6% FML, higher levels of dietary Met, i.e., 0.69 and 0.82%, respectively, were required to maximize shrimp BW. However, a complete withdrawal of FML was detrimental to shrimp BW even at 0.82% Met. In comparison, at 12 and 18% FML, a dietary Met content of only 0.58% was sufficient to enhance shrimp BW.

The reduction in the dietary inclusion of FML resulted in a significantly lower BW at harvest, especially with diets containing 0 and 6% FML. The highest BW was achieved when shrimp were fed 12 and 18% FML (*p* < 0.05), but the latter did not differ from 6% (*p* > 0.05). Similarly, reducing dietary Met content from 0.82 to 0.58 and 0.69% resulted in a significantly lower BW in *L. vannamei* (*p* < 0.05). However, no difference was detected in BW between shrimp fed 0.58 and 0.69% dietary Met (*p* > 0.05). At the lowest level of dietary Met, i.e., 0.58%, diets containing 12 and 18% FML outperformed the diets with 6 and 0% FML (*p* < 0.05). A similar but non-statistically significant trend was detected with higher levels of dietary Met, i.e., 0.69 and 0.82% (*p* > 0.05). 

The amount of dietary Met required to maximize shrimp BW was also a function of FML level. When FML was decreased to 6% or 0%, the highest BW was achieved with 0.69–0.82% Met, but at 12 and 18%, a minimum of 0.58% Met was sufficient. Thus, in general, as FML levels were challenged, higher levels of total dietary Met were required to maximize shrimp BW. Finally, shrimp fed the CTL achieved an equivalent BW compared to those fed 0 and 6% FML. Levels of FML higher than 6% regardless of dietary Met content resulted in significantly higher BW compared to the CTL (*p* < 0.05).

### 3.2. Dietary Protein and Amino Acid Digestibility

There was an increasing trend towards in apparent crude protein digestibility (ACPDC) with higher dietary inclusion levels of FML (Table 6). ACPDCs varied from a minimum of 81.1 ± 1.7% for shrimp fed a diet without FML to a high of 88.6 ± 2.1% for 18% FML. The apparent amino acid digestibility coefficients (AAADCs) followed a similar fashion for both EAAs and non-essential AAs (NEAAs). An exception was detected in ADCs for Lys, Met, and Met + Cys, which all fell within the same range in diets containing 0, 6, and 12% FML (91.0–91.8, 90.4–91.3, and 88.0–89.2%, respectively). However, higher ADCs values for these EAAs were found for the 18% FML diet (93.6, 93.2, and 91.6%, respectively). 

In indoor tanks, shrimp fed a diet deprived of FML achieved the highest final survival compared to other dietary treatments (Table 7, *p* < 0.005). Survival reached a mean of 87.5 ± 7.8% (*p* > 0.05). At harvest, shrimp fed 0% FML grew at 0.62 ± 0.04 g/week and achieved 12.69 ± 0.55 g final BW, both significantly lower than other dietary treatments (*p* < 0.05). However, no statistically significant differences were detected for gained shrimp yield (557 ± 80 g/m^2^) and FCR (2.68 ± 0.37). There was a significantly lower AFI when shrimp were fed 0 and 6% FML, but the later did not differ statistically from 12 and 18% FML (*p* > 0.05). 

### 3.3. Economic Efficiency

Total shrimp production cost, gross revenue, profit, and return on investment (ROI) were driven by FCR, yield, shrimp final BW, and formula costs, i.e., feed sales price. Formulation costs ranged from a minimum of 0.706 (0% FML with 0.56% Met) to a maximum of 0.943 USD/kg (18% FML with 0.82% Met; Table 1). The dietary inclusion of FML and total Met content both affected formula costs. The higher the dietary inclusion of FML, the higher the formula costs within the same level of dietary Met. Although a reduction in FML required higher levels of crystalline amino acid (CAA) supplementation, including DL-Met-Met, costs were offset by higher inclusions of SBM, which is a cheaper source of protein compared to FML. A reduction in FML from 18 to 12, 6%, and 0 at 0.69% dietary Met resulted in formula savings of 6.2, 13.3, and 23.1%, respectively. The increase in dietary Met content within the same level of FML also raised formula costs. However, the increase was less critical. Raising total Met content from 0.58 to 0.69 and 0.82% in diets deprived of FML impacted formula costs at 0.7 and 1.5%, respectively. A similar increase in formula cost was observed with 18% FML.

By taking into account the total production costs and shrimp performance data, it was possible to determine the ROI for each individual rearing tank and dietary treatment (Table 8). Economic analysis has indicated that FML levels significantly affected all parameters analyzed (*p* < 0.05), as opposed to the dietary levels of Met which showed no effect (*p* > 0.05). There was also no significant interaction between FML levels and dietary Met content for total production cost, gross revenue, profit, and ROI (*p* > 0.05). 

Production cost reduced progressively with lower dietary inclusions of FML, from a high of 4.10 ± 0.13 (18% FML) to a low of 3.24 ± 0.10 USD/kg (0% FML). There was on average a reduction in 0.86 USD/kg in production cost when FML was completely eliminated, i.e., from 18% to 0 FML. Although gross revenue was impacted with a reduction in FML, it was not significantly different between diets containing 6, 12, and 18% FML (*p* > 0.05). Gross revenue with 12% FML (4.78 ± 0.42 USD/kg) was higher than 0% FML (4.32 ± 0.28 USD/kg), but it did not differ from 6% (4.54 ± 0.41 USD/kg). On the other hand, the lowest profits were achieved with higher levels of FML, i.e., 12% (0.90 ± 0.38 USD/kg) and 18% (0.59 ± 0.28 USD/kg), although the former did not differ statistically from 0% (1.07 ± 0.27 USD/kg) and 6% FML (0.95 ± 0.30 USD/kg). 

Raising shrimp with a diet containing 18% FML led to the lowest ROI, at 14.3 ± 6.4%. Interestingly, the highest ROI was obtained with diets containing no or only 6% FML (33.2 ± 8.4 and 26.5 ± 7.9%, respectively). At no FML, ROI was significantly higher than 12 and 18% FML. At moderate levels of dietary inclusion, i.e., 6 and 12% FML, no differences in ROI could be observed. 

## 4. Discussion

This study has demonstrated that dietary FML levels and Met (Met + Cys) content and their interaction significantly impact final shrimp BW. The responses in BW as a function of FML level varied according to the dietary Met (Met + Cys) content. At the lowest dietary Met (Met + Cys), i.e., 0.58% (1.05%), we observed that FML could only be reduced from 18 to 12%. Further reductions led to a reduced shrimp BW at harvest. At moderate levels of dietary Met, i.e., 0.69% (1.16%), FML could be completely eliminated without any impact to shrimp BW, but with an adverse effect on yield. In comparison, at the highest dietary Met, i.e., 0.82% (1.29%), FML could be reduced from 18 to 6%, but complete withdrawal negatively impacted BW. Therefore, the levels of dietary Met (Met + Cys) required to maximize shrimp BW at 0 and 6% FML ranged between 0.69 (1.16) and 0.82% (1.29%), while at 12 and 18%, only 0.58% (1.05%) was needed. Notably, Met apparent digestibility coefficient was not affected by the FML across diets, which means that even on a digestibility basis this difference will hold true. Therefore, it is clear that an effective reduction in FML in diets for *L. vannamei* is dependent on appropriate dietary Met (Met + Cys) levels. This can be achieved through the supplementation with crystalline Met or by raising protein ingredients rich in these intact AAs. This is consistent with other work which indicates the importance of supplemental Met while challenging FML levels [20,42]. 

We found that under outdoor and indoor tank conditions, the optimal dietary FML level without an adverse effect on the growth performance of *L. vannamei* was reached at 6%. This finding is corroborated with the work of Suárez et al. [18], who raised juvenile *L. vannamei* (0.3 ± 0.1 g) under clear-water conditions for 95 days. The authors replaced FML with SBM while keeping the dietary levels of Met + Cys (1.0–1.2%) consistent with the use of canola meal, a natural source of sulfur-containing AAs. They found that a combination of SBM and canola meal was able to reduce FML from 30 to 6% but not beyond that. In the current study, to completely withdraw FML from the diets, SBM levels were increased to 46.08%. In this case, most of the dietary protein came from plant sources (wheat flour, wheat gluten) with only a small portion (4%) derived from animals (squid and krill). We found that even at higher levels of dietary SBM (45% of the diet), the ADCs for dietary protein and AAs were adequate (>80%). Thus, the reduced shrimp growth performance with 0% FML was likely driven by other factors, including a poorer feed attractability. We could not observe any negative response in AFI with the outdoor rearing system likely because shrimp were fed with feed restrictions. However, in the indoor systems, when shrimp were fed in excess, there was a reduced AFI when shrimp were fed diets containing 0 and 6% FML. 

In other work, plant-based feeds [6,7] and feeds containing in excess of 50% SBM have been advocated for juvenile *L. vannamei* [8,66,67,68,69]. Some of these studies report that a complete replacement of FML for SBM can be achieved with or without the use of proteins from terrestrial animals, such as poultry meal. These studies also adopted green-water conditions, but shrimp were raised with a much lower stocking density (25 shrimp/m^2^ [66]; 35 shrimp/m^2^ [7,8,67]; 37.5 shrimp/m^2^ [6] compared to the present work (100 shrimp/m^2^). With a lower density, the nutrient contribution of naturally available food sources to shrimp growth can be significant and can lead to much higher levels of FML replacement. Their formulas also combined other protein sources for FML replacement in addition to SBM. For example, Amaya et al. [6] were able to reduce FML from 9% to 0 by increasing SBM from 32.58 to 39.08%. All of their diets contained a fixed level of 16.0% poultry by-product meal, except one which only contained plant ingredients (SBM, sorghum meal, corn gluten meal, fermented corn solubles). Therefore, FML is more likely to be fully replaced when feed formulation can rely on more than one or on a combination of substitute protein sources. 

In the current study, from the economic point of view, eliminating FML was as competitive as including 6% FML, with both being more advantageous than 12 and 18% FML. The production costs of diets containing lower levels of FML exceeded the benefits of a higher revenue, resulting in greater profit and ROI. The cost of FML was the main driver for a higher production cost recorded for diets 12 and 18% FML. In comparison, total dietary Met content had no effect over economical parameters, including production costs. 

## 5. Conclusions

Results from the present study show that a correct balance of FML and dietary Met has a critical effect on whiteleg shrimp performance. Shrimp final survival, growth, FCR, and the dietary protein and amino acid digestibility were not negatively affected by a reduction or complete elimination of FML with a proper balance with CAAs, including Met. However, while gained shrimp yield is reduced when FML is withdrawn, dietary inclusion levels of 12% or higher leads to increased costs, which are not offset by higher revenues. In conclusion, the total amount of dietary Met needed to maximize shrimp growth performance depends on the amount of dietary FML. Higher amounts of supplemental DL-methionyl-DL-methionine reduces the reliance on FML. A total dietary supplementation of DL-Met-Met of 0.34% can reduce FML inclusion from 18 to 6% without any negative effect on shrimp performance. Overall, results indicate that the use of FML can be minimized or completely eliminated without major detrimental effects on shrimp performance, as long as methionine requirement is met with proper supplementation of CAAs. Feeds with 0 FML or with only 6% with levels of dietary Met (as-fed basis), i.e., 0.69 and 0.82%, respectively, deliver the highest shrimp growth performance, profit, and return on investment compared to diets with higher levels.

## Figures and Tables

**Figure 1 animals-13-00020-f001:**
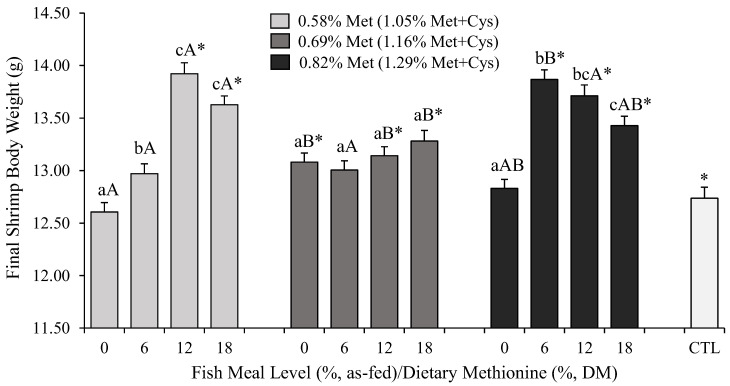
Mean (±standard error) body weight (g) of *L. vannamei* after 70 days of rearing in green water tanks of 1 m^3^. Different lowercase letters indicate statistically significant differences between fish meal (FML) levels within the same dietary methionine (Met) content at the α = 0.05 level according to Tukey’s HSD. Different capital letters refer to significant differences (*p* < 0.05) between Met levels within each FML dietary inclusion. Asterisks (*) indicate statistically significant differences between the commercial reference (CTL) and the experimental diets (*p* < 0.05, Student’s *t*-test).

**Table 1 animals-13-00020-t001:** Ingredient composition (%, as-is) and formula cost (USD/kg) of diets prepared to evaluate growth performance.

Ingredients (%, As-Is Basis)	Diets/Ingredient Composition (%, As-Is Basis)											
	0.58% Met (1.05% Met + Cys)				0.69% Met (1.16% Met + Cys)				0.82% Met (1.29% Met + Cys)			
% Fish Meal (FML) Level	0	6	12	18	0	6	12	18	0	6	12	18
Wheat flour ^1^	30.00	30.00	30.00	30.00	30.00	30.00	30.00	30.00	30.00	30.00	30.00	30.00
Soybean meal ^2^	46.08	37.80	28.95	20.73	46.08	37.80	28.95	20.73	46.08	37.80	28.95	20.73
Wheat gluten meal ^3^	5.00	5.00	5.00	5.00	5.00	5.00	5.00	5.00	5.00	5.00	5.00	5.00
Salmon meal ^4^	-	6.00	12.00	18.00	-	6.00	12.00	18.00	-	6.00	12.00	18.00
Cassava starch ^5^	0.26	1.61	4.46	6.20	0.13	1.49	4.34	6.09	-	1.35	4.20	5.95
Salmon oil	3.02	3.10	3.13	3.19	3.02	3.10	3.13	3.19	3.02	3.10	3.13	3.19
Soy lecithin	3.38	3.00	2.60	2.20	3.38	3.00	2.60	2.20	3.38	3.00	2.60	2.20
Yellow kaolin	-	2.00	4.00	5.00	-	2.00	4.00	5.00	-	2.00	4.00	5.00
Krill meal ^6^	2.00	2.00	2.00	2.00	2.00	2.00	2.00	2.00	2.00	2.00	2.00	2.00
Squid meal ^7^	2.00	2.00	2.00	2.00	2.00	2.00	2.00	2.00	2.00	2.00	2.00	2.00
MSP ^8^	1.45	1.39	1.39	1.39	1.45	1.39	1.39	1.39	1.45	1.39	1.39	1.39
L-Lysine ^9^	0.47	0.43	0.43	0.41	0.47	0.43	0.43	0.41	0.47	0.43	0.43	0.41
Magnesium sulphate	0.004	0.49	-	-	0.004	0.49	-	-	0.004	0.49	-	-
Calcium carbonate	1.97	1.00	-	-	1.97	1.00	-	-	1.97	1.00	-	-
Vitamin-mineral premix ^10^	1.00	1.00	1.00	1.00	1.00	1.00	1.00	1.00	1.00	1.00	1.00	1.00
Potassium chloride	1.13	0.99	0.84	0.68	1.13	0.99	0.84	0.68	1.13	0.99	0.84	0.68
L-Arginine ^11^	-	0.20	0.47	0.73	-	0.20	0.47	0.73	-	0.20	0.47	0.73
Synthetic binder ^12^	0.50	0.50	0.50	0.50	0.50	0.50	0.50	0.50	0.50	0.50	0.50	0.50
Salt	1.35	1.14	0.91	0.69	1.35	1.14	0.91	0.69	1.35	1.14	0.91	0.69
L-Threonine ^13^	0.12	0.14	0.14	0.14	0.12	0.14	0.14	0.14	0.12	0.14	0.14	0.14
L-Trytophan ^14^	0.04	0.06	0.09	0.11	0.04	0.06	0.09	0.11	0.04	0.06	0.09	0.11
DL-Met-Met ^15^	0.14	0.09	0.05	-	0.25	0.20	0.16	0.11	0.39	0.34	0.30	0.25
Cholesterol ^16^	0.07	0.04	0.02	-	0.07	0.04	0.02	-	0.07	0.04	0.02	-
Vitamin C ^17^	0.03	0.03	0.03	0.03	0.03	0.03	0.03	0.03	0.03	0.03	0.03	0.03
Formula cost (USD/kg)	0.756	0.822	0.878	0.932	0.761	0.827	0.883	0.937	0.767	0.833	0.889	0.943

^1^ 10.82% crude protein (CP), 0.18% methionine (Met), 0.41% methionine + cysteine (Met + Cys), 0.23% lysine (Lys), 0.28% threonine (Thr), 0.40% arginine (Arg). ^2^ Bunge Alimentos S.A. (Luiz Eduardo Magalhães, Brazil). 47.88% CP, 0.62% Met, 1.30% M + C, 2.91% Lys, 1.83% Thr, 3.45% Arg. ^3^ Amytex 100. Tereos Syral S.A.S. (Marckolsheim, France). 78.71% CP, 1.22% Met, 2.80% M + C, 1.37% Lys, 2.01% Thr, 2.84% Arg. ^4^ Pesquera Pacific Star S.A. (Puerto Montt, Chile). 63.61% CP, 1.61% Met, 2.17% M + C, 4.30% Lys, 2.47% Thr, 0.59% Arg. ^5^ 0.26% CP. ^6^ Qrill^TM^ Antarctic krill meal, Aker BioMarine Antarctic AS (Lysaker, Norway). 57.05% CP, 1.63% Met, 2.08% M + C, 3.92% Lys, 2.54% Thr, 3.45% Arg. ^7^ 70.66% CP, 2.34% Met, 3.17% M + C, 5.06% Lys, 3.33% Thr, 5.69% Arg. ^8^ Monosodium phosphate. 0.60% calcium, 20.70% phosphorous, 14.12% available phosphorous. ^9^ Biolys^®^, L-lysine, Evonik Operations GmbH (Hanau, Germany). 54.6% Lys. ^10^ Rovimix Camarao Intensivo. DSM Produtos Nutricionais Brasil Ltda. (São Paulo, Brazil). Guarantee levels per kg of product: vitamin A, 1,250,000 IU; vit. D3, 350,000 IU; vit. E, 25,000 IU; vit. K3, 500 mg; vit. B1, 5000 mg; vit. B2, 4000 mg; vit. B6; 10 mg; nicotinic acid, 15,000 mg; pantothenic acid, 10,000 mg; biotin, 150 mg; folic acid, 1250 mg; vit. C, 25,000 mg; choline, 50,000 mg; inositol, 20,000 mg; Fe 2000 mg; Cu, 3500 mg; chelated Cu, 1500 mg; Zn, 10,500 mg; chelated Zn, 4500 mg; Mn, 4000 mg; Se, 15 mg; chelated Se, 15 mg; I, 150 mg; Co, 30 mg; Cr, 80 mg; filler, 1000 g. ^11^ L-Arginine HCl, Sigma-Aldrich Co. (St. Louis, MO, USA), 98.5% Arg. ^12^ Nutri-Bind Aqua Veg Dry, Nutri-Ad International NV (Dendermonde, Belgium). Synthetic pellet binder consisting of calcium lignosulfonate (94.00%) and guar gum (6.00%). ^13^ ThreAMINO^®^, L-Threonine, Evonik Operations GmbH (Hanau, Germany). Min. 98.5% Thr. ^14^ TrypAMINO^®^, L-Tryptophan, Evonik Operations GmbH (Hanau, Germany). 98% tryptophan. ^15^ Aquavi^®^ Met-Met, DL-methionyl-DL-methionine, Evonik Operations GmbH (Hanau, Germany). 98% Met (min. 95% DL-Met-Met and max. 3% D, L-Met). ^16^ Cholesterol SF, Dishman Netherlands B.V. (Veenendaal, Netherlands). 91% cholesterol. ^17^ Rovimix^®^ Stay C^®^ 35, DSM Produtos Nutricionais Brasil Ltda. (São Paulo, Brazil). 350.0 g kg^−1^ phosphorylated vitamin C.

**Table 2 animals-13-00020-t002:** Proximate and amino acid composition (% of the diet, as-is) of experimental diets to evaluate growth performance.

Nutrient	Diets/Nutrient Composition (%, As-Is Basis)												
	0.58% Met (1.05% Met + Cys)				0.69% Met (1.16% Met + Cys)				0.82% Met (1.29% Met + Cys)				
% Fish Meal (FML) Level	0	6	12	18	0	6	12	18	0	6	12	18	CTL ^1^
Dry matter	90.35	88.75	89.02	88.80	90.11	90.48	88.88	87.73	90.13	90.17	88.85	88.98	90.47
Crude protein	34.57	34.20	34.36	34.94	34.40	35.21	34.41	34.39	34.03	34.87	34.39	35.07	39.25
Ether extract	7.62	7.90	8.89	8.11	7.45	7.88	7.52	7.21	7.34	7.48	7.67	7.63	6.90
Crude ash	8.96	10.96	11.45	11.90	8.90	10.39	10.89	11.86	9.28	10.61	11.23	12.24	12.17
Fiber	3.25	2.30	1.91	1.48	3.27	2.56	2.37	2.16	3.21	2.53	2.30	1.58	2.99
NFE ^2^	35.95	33.39	32.41	32.37	36.09	34.44	33.69	32.11	36.27	34.68	33.26	32.46	29.16
Gross energy (MJ/kg) ^3^	17.35	16.93	17.19	17.01	17.27	17.34	16.88	16.48	17.17	17.15	16.86	16.87	17.00
Essential amino acids (EAAs)													
Arginine	2.09	2.14	2.32	2.48	2.06	2.22	2.31	2.39	2.06	2.22	2.29	2.48	2.30
Histidine	0.75	0.73	0.73	0.73	0.76	0.76	0.72	0.71	0.76	0.76	0.74	0.72	1.02
Isoleucine	2.43	2.35	2.30	2.26	2.41	2.41	2.29	2.22	2.40	2.40	2.26	2.25	2.84
Leucine	2.43	2.35	2.30	2.26	2.41	2.41	2.29	2.22	2.40	2.40	2.26	2.25	2.84
Lysine	1.74	1.72	1.81	1.85	1.74	1.78	1.78	1.76	1.75	1.77	1.76	1.85	2.21
Methionine	0.57	0.56	0.59	0.62	0.67	0.71	0.71	0.69	0.80	0.83	0.83	0.83	0.86
Met + Cys	1.07	1.04	1.04	1.04	1.17	1.20	1.17	1.12	1.29	1.31	1.28	1.27	1.29
Phenylalanine	1.66	1.58	1.51	1.45	1.65	1.63	1.50	1.43	1.65	1.61	1.48	1.43	1.79
Threonine	1.29	1.29	1.31	1.33	1.29	1.33	1.30	1.29	1.29	1.33	1.29	1.31	1.41
Valine	1.52	1.48	1.46	1.47	1.49	1.51	1.46	1.44	1.49	1.50	1.45	1.45	1.93
Non-essential amino acids (NEAAs)													
Alanine	1.35	1.40	1.46	1.56	1.33	1.43	1.46	1.51	1.33	1.42	1.45	1.53	2.19
Cysteine	0.50	0.47	0.46	0.42	0.50	0.49	0.45	0.43	0.49	0.49	0.45	0.43	0.43
Glycine	1.39	1.53	1.69	1.88	1.38	1.57	1.69	1.83	1.38	1.57	1.68	1.86	2.60
Serine	1.60	1.55	1.50	1.47	1.61	1.58	1.50	1.43	1.61	1.59	1.48	1.47	1.68
Proline	2.17	2.06	2.16	2.18	2.08	2.20	2.13	2.12	2.08	2.16	2.15	2.15	2.22
Aspartic acid	3.15	2.97	2.85	2.76	3.11	3.08	2.85	2.69	3.11	3.05	2.83	2.72	3.47
Glutamic acid	7.05	6.61	6.48	6.27	6.86	6.88	6.43	6.16	6.85	6.86	6.41	6.23	5.84
Sum EAA ^4^	14.47	14.22	14.32	14.44	14.48	14.75	14.35	14.14	14.60	14.81	14.35	14.56	17.20
Sum NEAA	17.20	16.59	16.60	16.53	16.88	17.23	16.52	16.15	16.84	17.13	16.43	16.38	18.42
Sum EAA + NEAA	31.67	30.81	30.92	30.97	31.36	31.98	30.87	30.30	31.44	31.94	30.78	30.94	35.62

^1^ Commercial control. ^2^ Nitrogen free extract. Calculated by subtraction (dry matter − (crude protein + ether extract + crude fiber + ash)). ^3^ Gross energy (GE) given on a DM basis. Calculated as GE = (4143 + (56 × ether extract (DM)) + (15 × crude protein (DM)) − (44 × crude ash (DM))) × 0.0041868. ^4^ Tryptophan not included.

**Table 3 animals-13-00020-t003:** Ingredient composition (%, as-is) of diets used to evaluate digestibility.

Ingredients	Diets/Ingredient Composition (%, As-Is)			
% Fish Meal (FML) Level	0	6	12	18
Wheat flour ^1^	28.60	29.48	31.17	34.39
Soybean meal ^2^	45.02	37.32	28.84	20.02
Wheat gluten meal ^3^	5.00	5.00	5.00	5.00
Salmon meal ^4^	-	6.00	12.00	18.00
Kaolin	1.24	3.13	4.81	5.00
Salmon oil	3.41	3.32	3.35	3.39
Soy lecithin	3.28	2.88	2.47	2.05
Calcium carbonate	1.95	2.02	2.09	2.50
Krill meal ^5^	2.00	2.00	2.00	2.00
Squid meal ^6^	2.00	2.00	2.00	2.00
MSP ^7^	1.70	1.63	1.55	1.39
Vitamin-mineral premix ^8^	1.00	1.00	1.00	1.00
Potassium chloride	0.984	0.89	0.79	0.68
Salt	0.99	0.89	0.78	0.69
Chromic oxide III ^9^	0.50	0.50	0.50	0.50
Synthetic binder ^7^	0.50	0.50	0.50	0.50
L-Lysine ^7^	0.55	0.40	0.30	0.20
DL-Met-Met ^7^	0.41	0.34	0.27	0.20
Magnesium sulphate	0.57	0.37	0.17	-
L-Tryptophan ^7^	0.01	0.06	0.12	0.17
Cholesterol ^7^	0.10	0.10	0.10	0.10
L-Threonine ^7^	0.13	0.10	0.09	0.07
L-Arginine HCl ^7^		0.03	0.08	0.10
Vitamin C ^7^	0.04	0.04	0.04	0.04

^1^ 12.80% moisture, 11.44% crude protein (CP), 0.97% ether extract (EE), 0.19% crude fiber (CF), 0.68% ash, 0.18% methionine (Met), 0.27% lysine (Lys), 0.44% methionine + cysteine (M + C). ^2^ Bunge Alimentos S.A. (Luiz Eduardo Magalhães, Brazil). 10.30% moisture, 47.38% CP, 2.28% EE, 5.99% CF, 6.05% ash, 0.61% Met, 2.88% Lys, 1.28% M + C. ^3^ Amytex 100. Tereos Syral S.A.S. (Marckolsheim, France). 6.75% moisture, 79.68% CP, 2.44% EE, 0.41% CF, 1.87% ash, 1.16% Met, 1.35% Lys, 2.68% M + C. ^4^ Pesquera Pacific-Star (Puerto Montt, Chile). 10.89% moisture, 64.44% CP, 8.71% EE, 0.21% CF, 16.12% ash, 1.87% Met, 4.97% Lys, 2.70% M + C. ^5^ Qrill™ Antarctic krill meal, Aker BioMarine Antarctic AS (Lysaker, Norway). 6.61% moisture, 57.05% CP, 18.47% EE, 11.82% ash, 1.63% Met, 3.92% Lys, 2.08% M + C. ^6^ 9.75% moisture, 83.13% CP, 5.65% EE, 2.34% Met, 5.06% Lys, 3.17% M + C. ^7^ Check Table 1 for composition. ^8^ Vaccinar Industria e Comercio Ltda. (Pinhais, Brazil). Guarantee levels per kg of product: vitamin A, 1,200,000 IU; vit. D3, 200,000 IU; vit. E, 60,000 mg; vit. K3, 1000 mg; vit. B1, 2400 mg; vit. B2, 2400 mg; vit. B6, 6000 mg; vit. B12, 4 mg; nicotinic acid, 10,000 mg; pantothenic acid, 5200 mg; biotin, 20 mg; folic acid, 400 mg; vit. C, 30,000 mg; choline, 50,000 mg; inositol, 80,000 mg; Fe 26,000 mg; Cu, 2000 mg; Zn, 20,000 mg; Mn, 5000 mg; Se, 100 mg; I, 600 mg; Co, 105 mg; Cr, 60 mg. ^9^ Vetec Química Fina Ltda. (Rio de Janeiro, Brazil). Minimum of 99.0% of Cr_2_O_3._

**Table 4 animals-13-00020-t004:** Crude protein and amino acid composition (% of the diet, as-is) of experimental diets used in the digestibility assay.

Composition	Diets/Amino Acid Composition (%, As-Is)				
% Fish Meal (FML) Level	0	6	12	18	CV (%) ^1^
Dry matter	91.04	90.67	90.51	90.20	0.39
Crude protein	34.29	34.20	33.93	33.54	0.98
Essential amino acids (EAAs)					
Arginine	1.96	2.00	2.01	2.01	1.19
Histidine	0.68	0.70	0.71	0.72	2.59
Isoleucine	1.32	1.35	1.36	1.39	2.21
Leucine	2.24	2.27	2.30	2.31	1.43
Lysine	1.71	1.70	1.73	1.75	1.28
Methionine	0.80	0.81	0.81	0.82	1.34
Met + Cys	1.22	1.26	1.28	1.30	2.66
Phenylalanine	1.44	1.50	1.54	1.58	4.12
Threonine	1.23	1.24	1.24	1.25	0.58
Valine	1.47	1.48	1.45	1.45	0.83
Non-essential amino acids (NEAAs)					
Alanine	1.53	1.47	1.39	1.32	6.51
Cysteine	0.43	0.45	0.47	0.48	5.23
Glycine	1.87	1.70	1.54	1.35	13.68
Serine	1.43	1.48	1.52	1.53	2.92
Proline	2.15	2.12	2.09	2.06	1.82
Aspartic acid	2.68	2.82	2.91	2.99	4.69
Glutamic acid	6.22	6.38	6.51	6.64	2.81
Sum EAA ^2^	12.84	13.05	13.16	13.28	1.46
Sum NEAA	16.31	16.42	16.42	16.37	0.32
Sum EAA + NEAA	29.14	29.47	29.58	29.66	0.77

^1^ Coefficient of variation. ^2^ Tryptophan not included.

**Table 5 animals-13-00020-t005:** Final shrimp growth performance (mean ± SD) in outdoor 1 m^3^ tanks as a function of dietary methionine (Met) and fish meal (FML) inclusion levels. Shrimp of 1.00 ± 0.08 g were raised with 100 animals/m^2^ for 70 days. Asterisks (*) refer to statistically significant differences between the commercial control (CTL) and the experimental diets (*p* < 0.05, Student’s *t*-test). Common letters in each column indicate non-statistically significant differences between different dietary fish meal levels according to Tukey’s HSD test (*p* > 0.05).

Variable	% Met	Mean ± SD	Dietary Fish Meal Level (%, As-Is)				
			0	6	12	18	CTL
Final survival (%)	0.58	92.1 ± 4.08	93.3 ± 2.1	92.4 ± 2.0	91.7 ± 1.9	92.5 ± 1.7	89.7 ± 2.8
	0.69	94.2 ± 4.06	94.1 ± 1.9	95.5 ± 2.0	95.1 ± 2.6	92.6 ± 1.9	-
	0.82	92.5 ± 5.46	90.6 ± 3.4	88.2 ± 2.9	93.1 ± 1.4	94.4 ± 1.7	-
Mean ± SD		-	92.7 ± 5.52	92.0 ± 5.72	93.4 ± 4.37	93.2 ± 3.84	-
Gained yield (g/m^2^)	0.58	1127 ± 56	1077 ± 20	1098 ± 20	1177 ± 30 *	1162 ± 20 *	1044 ± 33 *
	0.69	1145 ± 45	1131 ± 7 *	1142 ± 19 *	1149 ± 26 *	1132 ± 26	-
	0.82	1135 ± 84	1062 ± 30	1123 ± 47	1177 ± 47	1168 ± 28 *	-
Mean ± SD		-	1090 ± 54 ^a^	1121 ± 68 ^ab^	1166 ± 66 ^b^	1151 ± 55 ^b^	-
Growth (g/week)	0.58	1.17 ± 0.09	1.10 ± 0.04	1.14 ± 0.05	1.23 ± 0.01	1.20 ± 0.03	1.12 ± 0.06
	0.69	1.16 ± 0.07	1.15 ± 0.03	1.14 ± 0.05	1.15 ± 0.04	1.17 ± 0.03	-
	0.82	1.17 ± 0.09	1.12 ± 0.01	1.22 ± 0.03	1.21 ± 0.06	1.18 ± 0.04	-
Mean ± SD		-	1.12 ± 0.06	1.17 ± 0.10	1.19 ± 0.08	1.18 ± 0.08	
AFI (g/shrimp)	0.58	13.4 ± 0.5	12.9 ± 0.2	13.4 ± 0.3	13.9 ± 0.2 *	13.4 ± 0.1 *	12.8 ± 0.2 *
	0.69	13.2 ± 0.5	13.3 ± 0.2	13.1 ± 0.3	13.3 ± 0.3	13.3 ± 0.1	-
	0.82	13.4 ± 0.5	13.2 ± 0.2	13.5 ± 0.3	13.5 ± <0.01 *	13.6 ± 0.2 *	-
Mean ± SD		-	13.1 ± 0.4	13.3 ± 0.7	13.5 ± 0.5	13.4 ± 0.5	
FCR	0.58	1.19 ± 0.04	1.20 ± 0.02	1.22 ± 0.01	1.18 ± 0.02	1.16 ± 0.01 *	1.23 ± 0.02 *
	0.69	1.15 ± 0.05	1.17 ± 0.02	1.15 ± 0.02 *	1.16 ± 0.03	1.17 ± 0.02	-
	0.82	1.19 ± 0.08	1.24 ± 0.05	1.21 ± 0.04	1.16 ± 0.05	1.16 ± 0.01	-
Mean ± SD		-	1.21 ± 0.08	1.19 ± 0.06	1.16 ± 0.06	1.16 ± 0.05	-
Two-Way ANOVA		Survival	Yield		Growth	AFI	FCR
FML		0.891	0.007; 0 < 12, 18%, 6 = 0%, 6 = 12, 18%		0.122	0.132	0.175
Met		0.221	0.880		0.558	0.367	0.253
FML x Met		0.551	0.451		0.574	0.502	0.538

**Table 6 animals-13-00020-t006:** Mean (±SD) apparent digestibility coefficient (ADC, %) of protein and amino acids in diets containing graded levels of fish meal.

Nutrient	Diets/Apparent Digestibility Coefficient (ADC, %)			
% Fish Meal (FML) Level	0	6	12	18
Crude protein	81.1 ± 1.7	83.1 ± 9.2	84.7 ± 1.3	88.6 ± 2.1
Essential amino acids (EAAs)				
Arginine	91.9 ± 1.2	92.7 ± 4.0	93.1 ± 0.8	94.8 ± 0.8
Histidine	87.4 ± 1.6	88.1 ± 6.5	88.9 ± 1.2	90.9 ± 1.6
Isoleucine	87.8 ± 0.9	89.0 ± 5.1	89.7 ± 1.5	92.0 ± 1.3
Leucine	87.5 ± 1.5	88.9 ± 6.1	89.6 ± 1.4	92.2 ± 1.3
Lysine	91.0 ± 1.3	91.8 ± 4.1	91.8 ± 1.0	93.6 ± 1.1
Methionine	91.1 ± 0.9	90.4 ± 4.5	91.3 ± 1.5	93.2 ± 0.7
Met + Cys	88.0 ± 1.4	88.2 ± 5.9	89.2 ± 1.6	91.6 ± 1.1
Phenylalanine	88.6 ± 1.2	89.4 ± 5.6	89.9 ± 1.3	92.3 ± 1.3
Threonine	85.4 ± 2.3	87.2 ± 6.7	87.8 ± 1.5	90.5 ± 1.6
Valine	85.7 ± 0.8	87.2 ± 5.9	88.1 ± 1.3	90.9 ± 1.7
Non-essential amino acids (NEAAs)				
Alanine	81.5 ± 1.4	84.6 ± 7.0	86.3 ± 0.3	89.9 ± 1.7
Cysteine	82.6 ± 2.1	84.4 ± 8.1	85.6 ± 1.7	88.5 ± 1.8
Glycine	80.5 ± 2.0	84.2 ± 7.8	86.2 ± 0.5	89.7 ± 1.7
Serine	87.0 ± 1.9	88.2 ± 6.1	88.8 ± 1.1	91.4 ± 1.4
Proline	87.4 ± 1.7	89.7 ± 5.3	90.4 ± 1.1	93.2 ± 1.3
Aspartic acid	86.4 ± 1.5	87.7 ± 5.9	88.2 ± 1.2	90.6 ± 1.7
Glutamic acid	92.2 ± 1.0	93.2 ± 3.5	93.6 ± 0.6	95.2 ± 0.8

**Table 7 animals-13-00020-t007:** Final shrimp growth performance (mean ± SD) in a clear-water recirculating indoor tank system as a function of dietary fish meal (FML) level. Shrimp were raised for 93 days with 70 shrimp/m^2^ in 60 L tanks to determine feed digestibility. Different letters in the same line indicate statistically significant differences according to Tukey’s HSD test at α = 0.05.

Performance Parameter	Dietary Fish Meal Level (%, As-Is)					One-Way ANOVA
	0	6	12	18	Mean ± SD	
Initial body weight (g)	4.41 ± 0.14	4.52 ± 0.21	4.46 ± 0.09	4.47 ± 0.07	4.47 ± 0.14	0.368
Final survival (%)	91.7 ± 7.6	86.0 ± 8.0	83.0 ± 5.9	88.0 ± 7.6	87.5 ± 7.8	0.068
Final body weight (g)	12.69 ± 0.55 ^a^	14.13 ± 1.29 ^b^	14.28 ± 0.81 ^b^	14.35 ± 0.49 ^b^	-	<0.0001
Growth (g/week)	0.62 ± 0.04 ^a^	0.72 ± 0.09 ^b^	0.74 ± 0.06 ^b^	0.74 ± 0.04 ^b^	-	<0.0001
Gained yield (g/m^2^)	540 ± 61	532 ± 84	553 ± 75	611 ± 86	557 ± 80	0.093
AFI (g/shrimp)	18.7 ± 1.3 ^a^	19.1 ± 1.3 ^ab^	20.6 ± 1.3 ^c^	20.3 ± 1.0 ^bc^	-	0.003
FCR	2.63 ± 0.32	2.76 ± 0.47	2.83 ± 0.30	2.53 ± 0.34	2.68 ± 0.37	0.289

**Table 8 animals-13-00020-t008:** Economic analysis of experimental diets as a function of fish meal (FML) and dietary methionine (Met) content. Common letters indicate non-statistically significant differences between dietary FML inclusion levels according to Tukey’s HSD test (*p* < 0.05).

Diets	Economic Analysis (USD/kg)			ROI (%) ^1^
	Production Cost	Gross Revenue	Profit	
0% FML	3.24 ± 0.10 ^a^	4.32 ± 0.28 ^a^	1.07 ± 0.27 ^a^	33.2 ± 8.4 ^a^
6% FML	3.59 ± 0.19 ^b^	4.54 ± 0.41 ^ab^	0.95 ± 0.30 ^a^	26.5 ± 7.9 ^ab^
12% FML	3.88 ± 0.13 ^c^	4.78 ± 0.42 ^b^	0.90 ± 0.38 ^ac^	23.1 ± 9.5 ^b^
18% FML	4.10 ± 0.13 ^d^	4.69 ± 0.38 ^ab^	0.59 ± 0.28 ^c^	14.3 ± 6.4 ^c^
0.58% Met	3.68 ± 0.61	4.55 ± 0.38	0.87 ± 0.41	24.2 ± 8.2
0.69% Met	3.64 ± 0.72	4.55 ± 0.32	0.91 ± 0.25	25.7 ± 11.4
0.82% Met	3.76 ± 0.85	4.62 ± 0.37	0.86 ± 0.42	23.1 ± 11.8
Two-Way ANOVA	Production Cost	Gross Revenue	Profit	ROI (%)
FML	<0.0001	0.011	0.001	<0.0001
Met	0.130	0.784	0.997	0.889
FML × Met	0.530	0.354	0.352	0.339

^1^ Return on investment.

## Data Availability

The data that support the findings of this study are available from the corresponding author upon reasonable request.

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
