# Peer review of "Optimal Levels of Fish Meal and Methionine in Diets for Juvenile Litopenaeus vannamei to Support Maximum Growth Performance with Economic Efficiency"

_animals, 2022, doi:10.3390/ani13010020_

Round 1

Reviewer 1 Report

Dear Editor,

This is a well written manuscript clearly portraying an experimentation on white shrimps concerning the use of  animal meal and methionine on the shrimp diet. The methodology followed is suitable and easily followed according to the text. Results are clearly reported. The discussion, although small, is valid according to the results observed. Some minor points only are discussed that do not essentially need any revision from my part.

Page 4, footnotes on Table 1. The composition of the different ingredients is given as footnotes in this table. They are quite descriptive but still essential. They can be added as supplementary material in a different table. The same applies to the footnotes in Table 3.

Author Response

REVIEWER 1:

This is a well written manuscript clearly portraying an experimentation on white shrimps concerning the use of animal meal and methionine on the shrimp diet. The methodology followed is suitable and easily followed according to the text. Results are clearly reported. The discussion, although small, is valid according to the results observed. Some minor points only are discussed that do not essentially need any revision from my part.

Q1.R1: Page 4, footnotes on Table 1. The composition of the different ingredients is given as footnotes in this table. They are quite descriptive but still essential. They can be added as supplementary material in a different table. The same applies to the footnotes in Table 3.

A1.R1: We thank the reviewer for the time and effort in revising the manuscript. In regards to the reviewer´s 1 suggestion in creating an additional Table to incorporate the basic nutrient composition of raw material used in feed formulations shown in footnotes of Tables 1 and Table 3, we think this will take too much space and may be difficult to organize. The specifications of some of the feed additives used, such as the vitamin and mineral premix, synthetic binder, MSP, etc., does not match the nutrient profile of raw materials (crude protein, methionine, methionine + cysteine, lysine). This means too many lines would need to be created in the table in order to clearly characterize each product used.

Reviewer 2 Report

The Manuscript titled ‘Optimal levels of fish meal and methionine in diets for juvenile Litopenaeus vannamei to support maximum growth performance with economic efficiency’ is a planned study that has novelty and adds to the current knowledge on nutritional aspects this commercially important shrimp species. The study evaluated the growth performance of juvenile L. vannamei and the economics of fish meal reduction with the dietary supplementation of  methionine. The methodology used for the present study is standard, results are interesting and data are well presented. The findings of the study has field applications and useful to develop lost cost nutritionally balanced feed. The manuscript has merit for publication subject to minor revision.

Specific queries and suggestions

·         What was the criteria for selection of fish meal level in the experimental feed (6%, 12%, 18%)?

·         Line no. 16. 0.82%(1.29???). Mention the unit for clarity

·         The methodology part is very elaborate which can be condensed

·         Line no. 106. The title ‘digestibility’ is not apt here as the given text deals more with experimental setup

·         Line no. 129. Mention the type of diets prepared and feed size.

·         How was the FCR calculated. Have you accounted the feed wastage?

·         Table 2. Why the control feed of much higher protein (39.25%) compared to treatment diets was used in the study?

·         Specify the ideal fish meal and methionine inclusion level in the feed for better growth and economic returns based on the study.

Decision suggested: Minor revision

Author Response

REVIEWER 2:

The Manuscript titled ‘Optimal levels of fish meal and methionine in diets for juvenile Litopenaeus vannamei to support maximum growth performance with economic efficiency’ is a planned study that has novelty and adds to the current knowledge on nutritional aspects this commercially important shrimp species. The study evaluated the growth performance of juvenile L. vannamei and the economics of fish meal reduction with the dietary supplementation of  methionine. The methodology used for the present study is standard, results are interesting and data are well presented. The findings of the study has field applications and useful to develop lost cost nutritionally balanced feed. The manuscript has merit for publication subject to minor revision.

Specific queries and suggestions

Q1.R2: What was the criteria for selection of fish meal level in the experimental feed (6%, 12%, 18%)?

A1.R2: We have taken 12% as the industry standard and adopted 50% above (18% fishmeal) and 50% below (6% fishmeal) this inclusion level.

Q2.R2: Line no. 16. 0.82%(1.29???). Mention the unit for clarity

A2.R1: Corrected;

Q3.R2: The methodology part is very elaborate which can be condensed

A3.R2: We have deleted lines 175 through 183 and line 233 through 241.

Q4.R2: Line no. 106. The title ‘digestibility’ is not apt here as the given text deals more with experimental setup

A4.R2. Subtitles “Growth performance” and “Digestibility” were deleted.

Q5.R2: Line no. 129. Mention the type of diets prepared and feed size.

A5.R2: We have mentioned the pellet size in line 249 (length and diameter). We have added the word “pelleted” to the following sentence: For the growth performance evaluation, three sets of “pelleted” diets…

Q6.R2: How was the FCR calculated. Have you accounted the feed wastage?

A6.R2: We explained the FCR calculations in lines 299-302. “FCR was calculated on a DM basis, by dividing the total inputs of feed (g, dry-matter basis, DM) delivered during the entire rearing period by the total harvested shrimp bio-mass (g, as-is basis) from each tank. There was no feed left-overs during the growth performance assay. In the digestibility assay, shrimp were fed in excess and any left-overs recovered from feeding trays were over-dried, weighed and subtracted from the total amount of dry feed delivered;

Q7.R2: Table 2. Why the control feed of much higher protein (39.25%) compared to treatment diets was used in the study?

A7.R2. In some countries, high protein diets are marketed as a high-performance feed to increase its market value.High protein diet (39-40%) used in the control mimics the current industry practice in the region. For the treatment diets, we used 34-35% CP as we demonstrated in our previous work (Nunes et al. 2019) that juvenile shrimp don't need more than 34% CP in the diets when balanced for amino acid profile. Similar work was also done in Vietnam in commercial ponds (Masagounder et al. 2022). Such diet produces better protein utilization as well as better economic performance.

Nunes AJP, Sabry-Neto H, Masagounder K. Crude protein in low-fish meal diets for juvenile Litopenaeus vannamei can be reduced through a well-balanced supplementation of essential amino acids. J World Aquacult Soc. 2019;1–15. https://doi.org/10.1111/jwas.12605

  1. Masagounder, T. T. T. Hien, T. L. C. Tu, N. V. Tien, C. N. Tien, L. Q. Viet, T. N. D. Khoa, T. N. Hai, and P. M. Duc. 2022 Use of supplemental amino acids in reducing dietary fish meal, crude protein level and improving dietary protein utilization in the grower phase of Whiteleg shrimp. World Aquaculture 2022, Nov 29-Dec 2, Singapore, published in abstract book: https://wasblobstorage.blob.core.windows.net/meeting-abstracts/WA2022AbstractBook.pdf

Q8.R2: Specify the ideal fish meal and methionine inclusion level in the feed for better growth and economic returns based on the study.

A8.R2: please check lines 531-533. “Feeds with zero FML or with only 6% with levels of dietary Met, i.e., 0.69 and 0.82%, respectively, delivers the highest shrimp growth performance, profit and return on in-vestment compared to diets with higher levels.”

Decision suggested: Minor revision